# Joint Perception and Control as Inference with an Object-based Implementation

## Abstract

Existing model-based reinforcement learning methods often study perception modeling and decision making separately. We introduce joint Perception and Control as Inference (PCI), a general framework to combine perception and control for partially observable environments through Bayesian inference. Based on the fact that object-level inductive biases are critical in human perceptual learning and reasoning, we propose Object-based Perception Control (OPC), an instantiation of PCI which manages to facilitate control using automatic discovered object-based representations. We develop an unsupervised end-to-end solution and analyze the convergence of the perception model update. Experiments in a high-dimensional pixel environment demonstrate the learning effectiveness of our object-based perception control approach. Specifically, we show that OPC achieves good perceptual grouping quality and outperforms several strong baselines in accumulated rewards.

## 1 Introduction

Human-like computing, which aims at endowing machines with human-like perceptual, reasoning and learning abilities, has recently drawn considerable attention (Lake, 2014; Lake *et al.*, 2015; Baker *et al.*, 2017). In order to operate within a dynamic environment while preserving homeostasis (Kauffman, 1993), humans maintain an internal model to learn new concepts efficiently from a few examples (Friston, 2005). The idea has since inspired many *model-based* reinforcement learning (MBRL) approaches to learn a concise perception model of the world (Kaelbling *et al.*, 1998). MBRL agents then use the perceptual model to choose effective actions. However, most existing MBRL methods separate perception modeling and decision making, leaving the potential connection between the objectives of these processes unexplored. A notable work by Hafner *et al.* (2020) provides a unified framework for perception and control. Built upon a general principle this framework covers a wide range of objectives in the fields of representation learning and reinforcement learning. However, they omit the discussion on combining perception and control for partially observable Markov decision processes (POMDPs), which formalizes many real-world decision-making problems. In this paper, therefore, we focus on the joint perception and control as inference for POMDPs and provide a specialized joint objective as well as a practical implementation.

Many prior MBRL methods fail to facilitate common-sense physical reasoning (Battaglia *et al.*, 2013), which is typically achieved by utilizing object-level *inductive biases*, e.g., the prior over observed objects' properties, such as the type, amount, and locations. In contrast, humans can obtain these inductive biases through interacting with the environment and receiving feedback throughout their lifetimes (Spelke *et al.*, 1992), leading to a unified hierarchical and behavioral-correlated perception model to perceive events and objects from the environment (Lee and Mumford, 2003). Before taking actions, a human agent can use this model to decompose a complex visual scene into distinct parts, understand relations between them, reason about their dynamics and predict the consequences of its actions (Battaglia *et al.*, 2013). Therefore, equipping MBRL with object-level inductive biases is essential to create agents capable of emulating human perceptual learning and reasoning and thus complex decision making (Lake *et al.*, 2015). We propose to train an agent in a similar way to gain inductive biases by learning the structured properties of the environment. This can enable the agent to plan like a human using its ability to think ahead, see what would happen for a range of possible choices, and make rapid decisions while learning a policy with the help of the inductive bias (Lake *et al.*, 2017). Moreover, in order to mimic a human's spontaneous acquisition of inductive biases

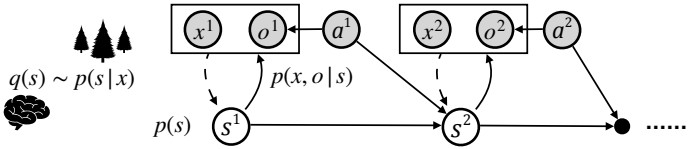

Figure 1: The graphical model of joint Perception and Control as Inference (PCI), where $s$ and $o$ represent the latent state and the binary optimality binary variable, respectively. The hierarchical perception model includes a bottom-up recognition model $q(s)$ and a top-down generative model $p(x, o, s)$ (decomposed into the likelihood $p(x, o|s)$ and the prior belief $p(s)$). Control is performed by taking an action $a$ to affect the environment state.

throughout its life, we propose to build a model able to acquire new knowledge online, rather than a one which merely generates static information from offline training (Dehaene *et al.*, 2017).

In this paper, we introduce joint Perception and Control as Inference (PCI) as shown in Fig. (1), a unified framework for decision making and perception modeling to facilitate understanding of the environment while providing a joint objective for both the perception and the action choice. As we argue that inductive bias gained in object-based perception is beneficial for control tasks, we then propose Object-based Perception Control (OPC), an instantiation of PCI which facilitates control with the help of automatically discovered representations of objects from raw pixels. We consider a setting inspired by real-world scenarios; we consider a partially observable environment in which agents' observations consist of a visual scene with compositional structure. The perception optimization of OPC is typically achieved by inference in a spatial mixture model through generalized expectation maximization (Dempster *et al.*, 1977), while the policy optimization is derived from conventional temporal-difference (TD) learning (Sutton, 1988). Proof of convergence for the perception model update is provided in Appendix A. We test OPC on the Pixel Waterworld environment. Our results show that OPC achieves good quality and consistent perceptual grouping and outperforms several strong baselines in terms of accumulated rewards.

## 2 RELATED WORK

**Connecting Perception and Control** Formulating RL as Bayesian inference over inputs and actions has been explored by recent works (Todorov, 2008; Kappen *et al.*, 2009; Rawlik *et al.*, 2010; Ortega and Braun, 2011; Levine, 2018; Tschiatschek *et al.*, 2018; Lee *et al.*, 2019b;a; Ortega *et al.*, 2019; Xin *et al.*, 2020; O'Donoghue *et al.*, 2020). The generalized free energy principle (Parr and Friston, 2019) studies a unified objective by heuristically defining entropy terms. A unified framework for perception and control from a general principle is proposed by Hafner *et al.* (2020). Their framework provides a common foundation from which a wide range of objectives can be derived such as representation learning, information gain, empowerment, and skill discovery. However, one trade-off for the generality of their framework is the loss in precision. Environments in many real-world decision-making problems are only partially observable, which signifies the importance of MBRL methods to solving POMDPs. However, relevant and integrated discussion is omitted in Hafner *et al.* (2020). In contrast, we focus on the joint perception and control as inference for POMDPs and provide a specialized joint-objective as well as a practical implementation.

**Model-based Deep Reinforcement Learning** MBRL algorithms have been shown to be effective in various tasks (Gu *et al.*, 2016), including operating in environments with high-dimensional raw pixel observations (Igl *et al.*, 2018; Shani *et al.*, 2005; Watter *et al.*, 2015; Levine *et al.*, 2016; Finn and Levine, 2017). Existing methods have considered incorporating reward structure into model-learning (Farahmand *et al.*, 2017; Oh *et al.*, 2017), while our proposed PCI takes one step forward by incorporating the perception model into the control-as-inference derivation to yield a single unified objective for multiple components in a pipeline. One of the methods closely related to OPC is the World Model (Ha and Schmidhuber, 2018), which consists of offline and separately trained models for vision, memory, and control. These methods typically produce entangled latent representations for pixel observations whereas, for real world tasks such as reasoning and physical interaction, it is often necessary to identify and manipulate multiple entities and their relationships for optimal performance. Although Zambaldi *et al.* (2018) has used the relational mechanism to discover and reason about entities, their model needs additional supervision of location information.

**Object-based Reinforcement Learning** The object-based approach, which recognizes decomposed objects from the environment observations, has attracted considerable attention in RL as well (Schmidhuber, 1992). However, most models often use pre-trained object-based representations rather than learning them from high-dimensional observations (Diuk *et al.*, 2008; Kansky *et al.*, 2017). When objects are extracted through learning methods, these models usually require supervised modeling of the object property, by either comparing the activation spectrum generated from neural network filters with existing types (Garnelo *et al.*, 2016) or leveraging the bounding boxes generated by standard object detection algorithms (Keramati *et al.*, 2018). MOREL (Goel *et al.*, 2018) applies optical flow in video sequences to learn the position and velocity information as input for model-free RL frameworks.

A distinguishing feature of our work in relation to previous works in MBRL and the object-based RL is that we provide the decision-making process with object-based abstractions of high-dimensional observations in an unsupervised manner, which contribute to faster learning.

**Unsupervised Object Segmentation** Unsupervised object segmentation and representation learning have seen several recent breakthroughs, such as IODINE (Greff *et al.*, 2019), MONet (Burgess *et al.*, 2019), and GENESIS (Engelcke *et al.*, 2020). Several recent works have investigated the unsupervised object extraction for reinforcement learning as well (Zhu *et al.*, 2018; Asai and Fukunaga, 2017; Kulkarni *et al.*, 2019; Watters *et al.*, 2019; Veerapaneni *et al.*, 2020). Although OPC is built upon a previous unsupervised object segmentation back-end (Greff *et al.*, 2017; van Steenkiste *et al.*, 2018), we explore one step forward by proposing a joint framework for perceptual grouping and decision-making. This could help an agent to discover structured objects from raw pixels so that it could better tackle its decision problems. Our framework also adheres to the Bayesian brain hypothesis by maintaining and updating a compact perception model towards the cause of particular observations (Friston, 2010).

## 3 METHODS

We start by introducing the environment as a partially observable Markov Decision Process (POMDP) with an object-based observation distribution in Sect. 3.1[1]. We then introduce PCI, a general framework for joint perception and control as inference in Sect. 3.2 and arrive at a joint objective for perception and control models. In the remainder of this section we propose OPC, a practical method to optimize the joint objective in the context of an object-based environment, which requires the model to exploit the compositional structure of a visual scene.

### 3.1 ENVIRONMENT SETTING

We define the environment as a POMDP represented by the tuple $\Gamma = \langle \mathcal{S}, \mathcal{P}, \mathcal{A}, \mathcal{X}, \mathcal{U}, \mathcal{R} \rangle$, where $\mathcal{S}, \mathcal{A}, \mathcal{X}$ are the state space, the action space, and the observation space, respectively. At time step $t$, we consider an agent's observation $\boldsymbol{x}^t \in \mathcal{X} \equiv \mathbb{R}^D$ as a visual image (a matrix of pixels) composited of $K$ objects, where each pixel $x_i$ is determined by exactly one object. The agent receives $\boldsymbol{x}^t$ following the conditional observation distribution $\mathcal{U}(\boldsymbol{x}^t|\boldsymbol{s}^t) : \mathcal{S} \to \mathcal{X}$, where the hidden state $\boldsymbol{s}^t$ is defined by the tuple $(\boldsymbol{z}^t, \boldsymbol{\theta}_1^t, \dots, \boldsymbol{\theta}_K^t)$. Concretely, we denote as $\boldsymbol{z}^t \in \mathcal{Z} \equiv [0,1]^{D \times K}$ the latent variable which encodes the unknown true pixel assignments, such that $z_{i,k}^t = 1$ iff pixel $z_i^t$ was generated by component $k$. Each pixel $x_i^t$ is then rendered by its corresponding object representations $\boldsymbol{\theta}_k^t \in \mathbb{R}^M$ through a pixel-wise distribution $\mathcal{U}_{\psi_{i,k}^t}(x_i^t|z_{i,k}^t = 1)$ [2], where $\psi_{i,k}^t = f_\phi(\boldsymbol{\theta}_k^t)_i$ is generated by feeding $\boldsymbol{\theta}_k^t$ into a differentiable non-linear function $f_\phi$. When the environment receives an action $a^t \in \mathcal{A}$, it moves to a new state $\boldsymbol{s}^{t+1}$ following the transition function $\mathcal{P}(\boldsymbol{s}^{t+1}|\boldsymbol{s}^t, a^t) : \mathcal{S} \times \mathcal{A} \to \mathcal{S}$. We assume the transition function could be parameterized and we integrate its parameter into $\phi$. To embed the control problem into the graphical model, we also introduce an additional binary random variable $o^t$ to represent the *optimality* at time step $t$, i.e., $o^t = 1$ denotes that time step $t$ is optimal, and $o^t = 0$ denotes that it is not optimal. We choose the distribution over $o^t$ to be $p(o^t = 1|\boldsymbol{s}^t, a^t) \propto \exp(r^t)$,

---

[1]Note that the PCI framework is designed for general POMDPs. We extend Sect. 3.1 to object-based POMDPs for the purpose of introducing the environment setting for OPC.

[2]We consider $\mathcal{U}$ as Gaussian, i.e., $\mathcal{U}_{\psi_{i,k}^t}(x_i^t|z_{i,k}^t = 1) \sim \mathcal{N}(x_i^t; \mu = \psi_{i,k}^t, \sigma^2)$ for some fixed $\sigma^2$.

where $r^t \in \mathbb{R}$ is the observed reward provided by the environment according to the reward function $\mathcal{R}(r^t|\boldsymbol{s}^t, a^t) : \mathcal{S} \times \mathcal{A} \to \mathbb{R}$. We denote the distribution over initial state as $p(\boldsymbol{s}^1) : \mathcal{S} \to [0, 1]$.

## 3.2 JOINT PERCEPTION AND CONTROL AS INFERENCE

To formalize the belief about the unobserved hidden cause of the history observation $\boldsymbol{x}^{\leq t}$, the agent maintains a perception model $q^w(\boldsymbol{s})$ to approximate the distribution over latent states as illustrated in Fig. (1). An agent's inferred belief about the latent state could serve as a sufficient statistic of the history and be used as input to its policy $\pi$, which guides the agent to act in the environment. The goal of the agent is to maximize the future optimality $o^{\geq t}$ while learning a perception model by inferring unobserved temporal hidden states given the past observations $\boldsymbol{x}^{\leq t}$ and actions $a^{\leq t}$, which is achieved by maximizing the following objective:

$$
\log p(o^{\geq t}, \boldsymbol{x}^{\leq t}|a^{<t}) = \log \sum_{\boldsymbol{s}^{\geq 1}, a^{\geq t}, \boldsymbol{x}^{>t}} p(o^{\geq t}, \boldsymbol{s}^{\geq 1}, \boldsymbol{x}^{\geq 1}, a^{\geq t}|a^{<t})
$$

$$
= \log \left[ \sum_{\boldsymbol{s}^{\leq t}} q^w \frac{\prod_{j=1}^{t} p(\boldsymbol{x}^j|\boldsymbol{s}^j) p(\boldsymbol{s}^1) \prod_{m=1}^{t-1} p(\boldsymbol{s}^{m+1}|\boldsymbol{s}^m, a^m)}{q^w} [①] \right], \quad (1)
$$

where we denote $q^w \doteq q(\boldsymbol{s}^{\leq t}|\boldsymbol{x}^{\leq t}, a^{<t})$ and use ① to represent the term related to control as

$$
① = \sum_{\boldsymbol{s}^{>t}, a^{\geq t}, \boldsymbol{x}^{>t}} \prod_{n=t} p(o^n|\boldsymbol{s}^n, a^n) \prod_{k=t} p(\boldsymbol{x}^{k+1}|\boldsymbol{s}^{k+1}) p(\boldsymbol{s}^{k+1}|\boldsymbol{s}^k, a^k) p(a^k|a^{<t}).
$$

The full derivation is presented in Appendix C.1. We denote $q^w \doteq q(\boldsymbol{s}^{\leq t}|\boldsymbol{x}^{\leq t}, a^{<t})$ and assume $q^w = p(\boldsymbol{s}^1) \prod_{g=2}^{t} q(\boldsymbol{s}^g|\boldsymbol{x}^{<g}, a^{<g})$, where we slightly abuse notation for $q^w$ by ignoring the fact that we sample from the model $p(\boldsymbol{s}^1)$ for $t = 1$. We then apply Jensen's inequality to Eq.(1) and get

$$
\log p(o^{\geq t}, \boldsymbol{x}^{\leq t}|a^{<t}) \geq \mathbb{E}_{\boldsymbol{s}^{\leq t} \sim q^w} \left[ \log \frac{\prod_{j=1}^{t} p(\boldsymbol{x}^j|\boldsymbol{s}^j) p(\boldsymbol{s}^1) \prod_{m=1}^{t-1} p(\boldsymbol{s}^{m+1}|\boldsymbol{s}^m, a^m)}{p(\boldsymbol{s}^1) \prod_{g=2}^{t} q(\boldsymbol{s}^g|\boldsymbol{x}^{<g}, a^{<g})} \right] + \mathbb{E}_{\boldsymbol{s}^{\leq t} \sim q^w} \left[ \log ① \right]
$$

$$
\underbrace{= \mathbb{E}_{\boldsymbol{s}^{\leq t} \sim q^w} \left[ \log \prod_{j=1}^{t} p(\boldsymbol{x}^j|\boldsymbol{s}^j) \right] - D_{KL} \left[ \prod_{g=1}^{t-1} q(\boldsymbol{s}^{g+1}|\boldsymbol{x}^{\leq g}, a^{\leq g}) \| p(\boldsymbol{s}^{g+1}|\boldsymbol{s}^g, a^g) \right]}_{\mathcal{L}^w(q^w, \phi)} + \mathbb{E}_{\boldsymbol{s}^{\leq t} \sim q^w} \left[ \log ① \right],
$$

where $D_{KL}$ represents the Kullback–Leibler divergence (Kullback and Leibler, 1951). As introduced in Sect. 3.1, we parameterize the transition distribution $p_\phi(\boldsymbol{s}^{t+1}|\boldsymbol{s}^t, a^t)$ and the observation distribution $p_\phi(\boldsymbol{x}^t|\boldsymbol{s}^t)$ by $\phi$. An instantiation to optimize the above evidence lower bound (ELBO) $\mathcal{L}^w(q^w, \phi)$ in an environment with explicit physical properties and high-dimensional pixel-level observations will be discussed in Sect. 3.3.

We now derive the control objective by extending ① as

$$
\log \sum_{\boldsymbol{s}^{>t}, a^{\geq t}, \boldsymbol{x}^{>t}} q(\boldsymbol{s}^{>t}, a^{\geq t}, \boldsymbol{x}^{>t}) \frac{\prod_{n=t} p(o^n|\boldsymbol{s}^n, a^n) \prod_{k=t} p_\phi(\boldsymbol{x}^{k+1}|\boldsymbol{s}^{k+1}) p_\phi(\boldsymbol{s}^{k+1}|\boldsymbol{s}^k, a^k) p(a^k|a^{<t})}{q(\boldsymbol{s}^{>t}, a^{\geq t}, \boldsymbol{x}^{>t})}
$$

where we assume $q(\boldsymbol{s}^{>t}, a^{\geq t}, \boldsymbol{x}^{>t}) = \prod_{h=t} p_\phi(\boldsymbol{x}^{h+1}|\boldsymbol{s}^{h+1}) p_\phi(\boldsymbol{s}^{h+1}|\boldsymbol{s}^h, a^h) \pi(a^h|\boldsymbol{s}^h)$ and denote $q^c \doteq q(\boldsymbol{s}^{>t}, a^{\geq t}, \boldsymbol{x}^{>t})$. We then apply Jensen's inequality to $\mathbb{E}_{\boldsymbol{s}^{\leq t} \sim q^w} [\log ①]$ and get

$$
\mathbb{E}_{\boldsymbol{s}^{\leq t} \sim q^w} [\log ①] \geq \mathbb{E}_{\boldsymbol{s}^{\leq t} \sim q^w} \left[ \sum_{\boldsymbol{s}^{>t}, a^{\geq t}, \boldsymbol{x}^{>t}} q^c \log \frac{\prod_{n=t} p(o^n|\boldsymbol{s}^n, a^n) \prod_{k=t} p_\phi(\boldsymbol{x}^{k+1}|\boldsymbol{s}^{k+1}) p_\phi(\boldsymbol{s}^{k+1}|\boldsymbol{s}^k, a^k) p(a^k|a^{<t})}{\prod_{h=t} p_\phi(\boldsymbol{x}^{h+1}|\boldsymbol{s}^{h+1}) p_\phi(\boldsymbol{s}^{h+1}|\boldsymbol{s}^h, a^h) \pi(a^h|\boldsymbol{s}^h)} \right]
$$

$$
= \mathbb{E}_{\boldsymbol{s}^{\leq t} \sim q^w} \left[ \sum_t \mathbb{E}_{\boldsymbol{s}^{t+1}, a^t, \boldsymbol{x}^{t+1} \sim \rho_\phi \pi} \left[ R(\boldsymbol{s}^t, a^t) - D_{KL}(\pi(a^t|\boldsymbol{s}^t) \| p(a^t|a^{<t})) \right] \right],
$$
$$(2)$$

where we denote $\rho_\phi$ as $p_\phi(\boldsymbol{x}^{h+1}|\boldsymbol{s}^{h+1}) p_\phi(\boldsymbol{s}^{h+1}|\boldsymbol{s}^h, a^h)$. Note that as this control objective is an expectation under $\rho_\phi$, reward maximization also bias the learning of our perception model. We will propose an option to optimize the control objective Eq.(2) in Sect. 3.4.

### 3.3 OBJECT-BASED PERCEPTION MODEL UPDATE

We now introduce OPC, an instantiation of PCI in the context of an object-based environment. Following the generalized expectation maximization (Dempster et al., 1977), we optimize the ELBO $\mathcal{L}^w(q^w, \phi)$ by improving the perception model $q^w$ about the true posterior $p_\phi(s^{t+1}|x^{t+1}, s^t, a^t)$ with respect to a set of object representations $\boldsymbol{\theta}^t = [\boldsymbol{\theta}_1^t, \ldots, \boldsymbol{\theta}_K^t] \in \Omega \subset \mathbb{R}^{M \times K}$.

**E-step to compute a new estimate $q^w$ of the posterior.** Assume the ELBO is already maximized with respect to $q^w$ at time step $t$, i.e., $q^w = p_\phi(s_i^{t+1}|x_i^{t+1}, s^t, a^t) = p_\phi(z_i^{t+1}|x_i^{t+1}, \boldsymbol{\psi}_i^t, a^t)$, we can generate a soft-assignment of each pixel to one of the $K$ objects as

$$\eta_{i,k}^t \doteq p_\phi(z_{i,k}^{t+1} = 1|x_i^{t+1}, \boldsymbol{\psi}_i^t, a^t). \tag{3}$$

**M-step to update the model parameter $\phi$.** We then find $\boldsymbol{\theta}^{t+1}$ to maximize the ELBO as

$$\boldsymbol{\theta}^{t+1} = \arg\max_{\boldsymbol{\theta}^{t+1}} \mathbb{E}_{z^{t+1} \sim \eta^t} \left[ \log p_\phi(x^{t+1}, z^{t+1}, \psi^{t+1}|s^t, a^t) \right] \doteq \arg\max_{\boldsymbol{\theta}^{t+1}} \Lambda_{\boldsymbol{\theta}^t}(\boldsymbol{\theta}^{t+1}). \tag{4}$$

See derivation in Appendix C.2. Note that Eq. (4) returns a set of points that maximize $\Lambda_{\boldsymbol{\theta}^t}(\boldsymbol{\theta}^{t+1})$, and we choose $\boldsymbol{\theta}^{t+1}$ to be any value within this set. To update the perception model by maximizing the evidence lower bound with respect to $q(s^t)$ and $\boldsymbol{\theta}^{t+1}$, we compute Eq. (3) and Eq. (4) iteratively. However, an analytical solution to Eq. (4) is not available because we use a differentiable non-linear function $f_\phi$ to map from object representations $\boldsymbol{\theta}_k^t$ into $\psi_{i,k}^t = f_\phi(\boldsymbol{\theta}_k^t)_i$. Therefore, we get $\boldsymbol{\theta}_k^{t+1}$ by

$$\boldsymbol{\theta}_k^{t+1} = \boldsymbol{\theta}_k^t + \alpha \left. \frac{\partial \Lambda_{\boldsymbol{\theta}^{t+1}}(\boldsymbol{\theta}^{t+2})}{\partial \boldsymbol{\theta}_k^{t+1}} \right|_{\boldsymbol{\theta}_k^{t+1} = \boldsymbol{\theta}_k^t} = \boldsymbol{\theta}_k^t + \alpha \sum_{i=1}^{D} \eta_{i,k}^t \cdot \frac{\psi_{i,k}^t - x_i^{t+1}}{\sigma^2} \cdot \frac{\partial \psi_{i,k}^t}{\partial \boldsymbol{\theta}_k^t}, \tag{5}$$

where $\alpha$ is the learning rate (see details of derivation in Appendix C.3).

We regard the iterative process as a $t_{ro}$-step rollout of $K$ copies of a recurrent neural network with hidden states $\boldsymbol{\theta}_k^t$ receiving $\eta_k^t \odot (\psi_k^t - x^{t+1})$ as input (see the inner loop of Algorithm 1). Each copy generates a new $\psi_k^{t+1}$, which is then used to re-estimate the soft-assignments $\eta_k^{t+1}$. We parameterize the Jacobian $\partial \psi_k^t / \partial \boldsymbol{\theta}_k^t$ and the differentiable non-linear function $f_{\phi_k}$ using a convolutional encoder-decoder architecture with a recurrent neural network bottleneck, which linearly combines the output of the encoder with $\boldsymbol{\theta}_k^t$ from the previous time step. To fit the statistical model $f_\phi$ to capture the regularities corresponding to the observation and the transition distribution for given POMDPs, we back-propagate the gradient of $\mathbb{E}_{\tau \sim \pi} \left[ \Lambda_{\boldsymbol{\theta}^t}(\boldsymbol{\theta}^{t+1}) \right]$ through "time" (also known as the BPTT Werbos (1988)) into the weights $\phi$. We demonstrate the convergence of a sequence of $\{\mathcal{L}_{\boldsymbol{\theta}^t}^w(q^w, \phi)\}$ generated by the perception update in Appendix A. The proof is presented by showing that the learning process follows the Global Convergence Theorem (Zangwill, 1969).

Our implementation of the perceptual model is based on the unsupervised perceptual grouping method proposed by (Greff et al., 2017; van Steenkiste et al., 2018). However, using other advanced unsupervised object representations learning methods as the perceptual model (such as IODINE (Greff et al., 2019) and MONet (Burgess et al., 2019)) is a straightforward extension.

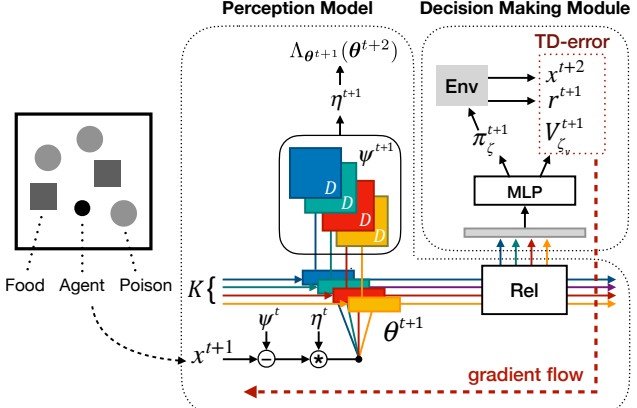

Figure 2: Illustration of OPC and the Pixel Waterworld environment. We regard the perception update as $K$ copies of an RNN with hidden states $\boldsymbol{\theta}_k^t$ receiving $\eta_k^t \odot (\psi_k^t - x^{t+1})$ as input. Each copy generates a new $\psi_k^{t+1}$, which is used to re-estimate the soft-assignments $\eta_k^{t+1}$ for the calculation of the expected log-likelihood $\Lambda_{\boldsymbol{\theta}^{t+1}}(\boldsymbol{\theta}^{t+2})$. The hidden states are then used as the input to decision-making (before being fed into an optional relational module) to guide the agent's action.

---

**Algorithm 1** Learning the Object-based Perception Control

---

Initialize $\boldsymbol{\theta}, \boldsymbol{\eta}, \phi, \zeta, \zeta_v, T_{epi}, t_{ro}, K$
Create $N_e$ environments that will execute in parallel
**while** training not finished **do**
    Initialize the history $\mathcal{D}_a, \mathcal{D}_x, \mathcal{D}_{\boldsymbol{\eta}}$ with environment rollouts for $t_{ro}+1$ time-steps under current policy $\pi_\zeta$
    **for** $T = 1$ to $T_{epi} - t_{ro}$ **do**
        $d\phi \leftarrow 0$
        Get $\boldsymbol{\eta}^{T-1}$ from $\mathcal{D}_{\boldsymbol{\eta}}$
        **for** $t = T$ to $T + t_{ro}$ **do**
            Get $a^t, \boldsymbol{x}^t$ from $\mathcal{D}_a, \mathcal{D}_x$ respectively
            Feed $\boldsymbol{\eta}_k^{t-1} \odot (\boldsymbol{\psi}_k^{t-1} - \boldsymbol{x}^t)$ into each of the $K$ RNN copy to get $\boldsymbol{\theta}_k^t$ and forward-output $\boldsymbol{\psi}_k^t$
            Compute $\boldsymbol{\eta}_k^t$ by Eq. 3
            $d\phi \leftarrow d\phi + \partial\left(-\Lambda_{\boldsymbol{\theta}^t}(\boldsymbol{\theta}^{t+1})\right)/\partial\phi$ by Eq. (4)
        Perform $a^{T+t_{ro}}$ according to policy $\pi_\zeta(a^{T+t_{ro}}|\boldsymbol{\theta}^{T+t_{ro}})$
        Receive reward $r^{T+t_{ro}}$ and new observation $\boldsymbol{x}^{T+t_{ro}+1}$
        Store $a^{T+t_{ro}}, \boldsymbol{x}^{T+t_{ro}+1}, \boldsymbol{\eta}^T$ in $\mathcal{D}_a, \mathcal{D}_x, \mathcal{D}_{\boldsymbol{\eta}}$ respectively
        Feed $\boldsymbol{\eta}_k^{T+t_{ro}} \odot (\boldsymbol{\psi}_k^{T+t_{ro}} - \boldsymbol{x}^{T+t_{ro}+1})$ into each of the $K$ RNN copy to get $\boldsymbol{\theta}_k^{T+t_{ro}+1}$
        $y \leftarrow r^{T+t_{ro}} + \gamma V_{\zeta_v}(\boldsymbol{\theta}^{T+t_{ro}+1})$
        $d\zeta \leftarrow \nabla_\zeta \log \pi_\zeta(a^{T+t_{ro}}|\boldsymbol{\theta}^{T+t_{ro}})(y - V_{\zeta_v}(\boldsymbol{\theta}^{T+t_{ro}}))$
        $d\zeta_v \leftarrow \partial\left(y - V_{\zeta_v}(\boldsymbol{\theta}^{T+t_{ro}})\right)^2/\partial\zeta_v$
        $d\phi \leftarrow d\phi + \partial\left(y - V_{\zeta_v}(\boldsymbol{\theta}^{T+t_{ro}})\right)^2/\partial\phi$
    Perform synchronous update of $\phi$ using $d\phi$, of $\zeta$ using $d\zeta$, and of $\zeta_v$ using $d\zeta_v$

---

### 3.4 DECISION-MAKING MODULE UPDATE

Recall that the control objective in Eq.(2) is the sum of the expected total reward along trajectory $\rho_\theta\pi$ and the negative KL divergence between the policy and our prior about the future actions. The prior can reflect a preference over policies. For example, a uniform prior in SAC (Haarnoja *et al.*, 2018) corresponds to the preference for a simple policy with minimal assumptions. This could prevent early convergence to sub-optimal policies. However, as we focus on studying the benefit of combing perception with control, we do not pre-impose a preference for policies. Therefore we set the prior equal to our training policy, resulting in a zero KL divergence term in the objective.

To maximize the total reward along trajectory $\rho_\theta\pi$, we follow the conventional temporal-difference (TD) learning approach (Sutton, 1988) by feeding the object abstractions into a small multilayer perceptron (MLP) (Rumelhart *et al.*, 1986) to produce a $(\dim_{\mathbb{R}}(\mathcal{A}) + 1)$-dimensional vector, which is split into a $\dim_{\mathbb{R}}(\mathcal{A})$-dimensional vector of $\pi_\zeta$'s (the 'actor') logits, and a baseline scalar $V_{\zeta_v}$ (the 'critic'). The $\pi_\zeta$ logits are normalized using a softmax function, and used as the multinomial distribution from which an action is sampled. The $V_{\zeta_v}$ is an estimate of the state-value function at the current state, which is given by the last hidden state $\boldsymbol{\theta}$ of the $t_{ro}$-step RNN rollout. On training the decision-making module, the $V_{\zeta_v}$ is used to compute the temporal-difference error given by

$$\mathcal{L}_{TD} = (y^{t+1} - V_{\zeta_v}(\boldsymbol{\theta}^{t+1}))^2, y^{t+1} = r^{t+1} + \gamma V_{\zeta_v}(\boldsymbol{\theta}^{t+2}), \tag{6}$$

where $\gamma \in [0, 1)$ is a discount factor. $\mathcal{L}_{TD}$ is used both to optimize $\pi_\zeta$ to generate actions with larger total rewards than $V_{\zeta_v}$ predicts by updating $\zeta$ with respect to the policy gradient

$$\nabla_\zeta \log \pi_\zeta(a^{t+1}|\boldsymbol{\theta}^{t+1})(y - V_{\zeta_v}(\boldsymbol{\theta}^{t+1})),$$

and to optimize $V_{\zeta_v}$ to more accurately estimate state values by updating $\zeta_v$. Also, differentiating $\mathcal{L}_{TD}$ with respect to $\phi$ enables the gradient-based optimizers to update the perception model. We provide the pseudo-code for one-step TD-learning of the proposed model in Algorithm 1. By grouping objects concerning the reward, our model distinguishes objects with visual similarities but different semantics, thus helping the agent to better understand the environment.

## 4 EXPERIMENTS

### 4.1 PIXEL WATERWORLD

We demonstrate the advantage of unifying decision making and perception modeling by applying OPC on an environment similar to the one used in COBRA (Watters *et al.*, 2019), a modified

Waterworld environment (Karpathy, 2015), where the observations are $84 \times 84$ grayscale raw pixel images composited of an agent and two types of randomly moving targets: the poison and the food, as illustrated in Fig. (2). The agent can control its velocity by choosing from four available actions: to apply the thruster to the left, right, up and down. A negative cost is given to the agent if it touches any poison target, while a positive reward for making contact with any food target. The optimal strategy depends on the number, speed, and size of objects, thus requiring the agent to infer the underlying dynamics of the environment within a given amount of observations.

The intuition of this environment is to test whether the agent can quickly learn dynamics of a new environment online without any prior knowledge, i.e., the execution-time optimization throughout the agent's life-cycle to mimic humans' spontaneous process of obtaining inductive biases. We choose Advantage Actor-Critic (A2C) (Mnih *et al.*, 2016) as the decision-making module of OPC without loss of generality, although other online learning methods are also applicable. For OPC, we use $K = 4$ and $t_{ro} = 20$ except for Sect. 4.5, where we analyze the effect of the hyper-parameter setting.

## 4.2 ACCUMULATED REWARD COMPARISONS

To verify object-level inductive biases facilitating decision making, we compare OPC against a set of baseline algorithms, including: 1) the standard A2C, which uses convolutional layers to transform the raw pixel observations to low dimensional vectors as input for the same MLP described in Sect. 3.4, 2) the World Model (Ha and Schmidhuber, 2018) with A2C for control (WM-A2C), a state-of-the-art model-based approach, which separately learns a visual and a memorization model to provide the in-

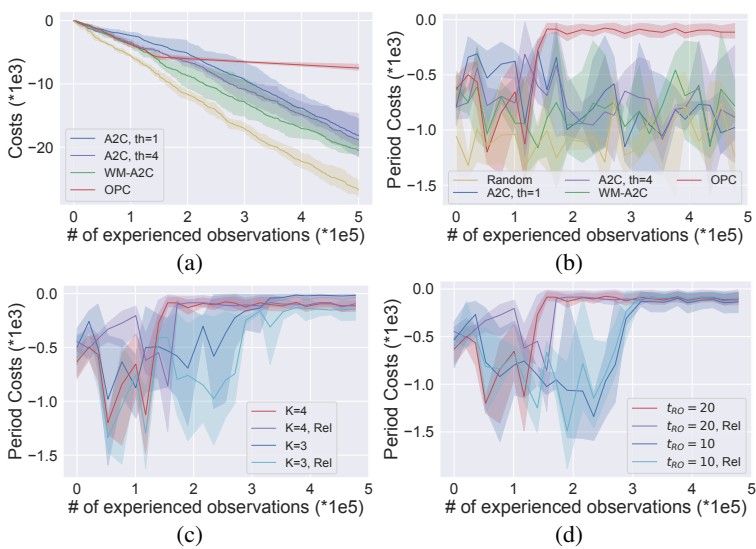

Figure 3: Performance comparisons between methods.

put for a standard A2C, and 3) the random policy. For both the baseline A2C, WM-A2C and the decision-making module of OPC, we follow the convention of Mnih *et al.* (2016) by running the algorithm in the forward view and using the same mix of $n$-step returns to update both $\pi_\zeta$ and $V_{\zeta_v}$. Details of the model and of hyperparameter setting can be found in Appendix B. We build the environment with two poison objects and one food object, and set the size of the agent $1.5$ times smaller than the target. Results are reported with the average value among separate runs of three random seeds. Note that the training procedure of WM-A2C includes independent off-line training for each component, thus requiring many more observation samples than OPC. Following its original early-stopping criteria, we report that WM-A2C requires 300 times more observation samples for separately training a perception model before learning a policy to give the results presented in Fig. (3).

Fig. (3a) shows the result of accumulated rewards after each agent has experienced the same amount of observations. Note that OPC uses the plotted number of experienced observations for training the entire model, while WM-A2C uses the amount of observations only for training the control module. It is clear that the agent with OPC achieves the greatest performance, outperforming all other agents regardless of model in terms of accumulated reward. We believe this advantage is due to the object-level inductive biases obtained by the perception model (compared to entangled representations extracted by the CNN used in standard A2C), and the unified learning of perception and control of OPC (as opposed to the separate learning of different modules in WM-A2C).

To illustrate each agent's learning process through time, we also present the period reward, which is the accumulated reward during a given period of environment interactions ($2e4$ of experienced

observations). As illustrated in Fig. (3b), OPC significantly improves the sample efficiency of A2C, which enables the agent acting in the environment to find an optimal strategy more quickly than agents with baseline models. We also find that the standard version of A2C with four parallel threads gives roughly the same result as the single-threaded version of A2C (the same as the decision-making module of OPC), eliminating the potential drawback of single-thread learning.

## 4.3 PERCEPTUAL GROUPING RESULTS

To demonstrate the performance of the perception model, we provide an example rollout at the early stage of training in Fig. (4). We assign a different color to each $\psi_k$ for better visualization and show through the soft-assignment $\eta$ that all objects are grouped semantically: the agent in blue, the food in green, and both poisons in red. During the joint training of perception and policy, the soft-assignment gradually constitutes a semantic segmentation as the TD signal improves the recognition when grouping pixels into different types based on the interaction with the environment. Consequently, the learned object representations $\theta$ becomes a semantic interpretation of raw pixels grouped into perceptual objects.

Figure 4: A sample rollout of perceptual grouping by OPC. The observation (top row), the next-step prediction $\psi_k$ of each copy of the $K$ RNN copy (rows 4 to 7), the $\sum_k \psi_k$ (row 2), and the soft-assignment $\eta$ of the pixels to each of the copies (row 3). We assign a different color to each $\psi_k$ for better visualization. This sample rollout shows that all objects are grouped semantically.

## 4.4 BENEFITS OF JOINT INFERENCE

To gain insights into the benefit of the control objective for joint inference, we further compare OPC against OP, a perception model with the same architecture as OPC but no guidance of the TD signal from RL, i.e., using the same unsupervised object segmentation back-end as in Greff et al. (2017). Fig. (5) shows the result of accumulated rewards and the soft-assignment $\eta$ produced by both perception models. Results are produced by models running in environments with the same random seed.

As illustrated in Fig. (5a), OPC outperforms OP in terms of the accumulated rewards through time with a given amount of observations. This performance difference owes to the fact shown in Fig. (5b), where the $\eta$ of OP are overlaid, and the shapes are not delineated in bold colors but are a mixture, showing that OP has not learned to segment the objects clearly. The pixel assignment (coloring) does not appear to converge even after an extremely large number of iterations, suggesting that OP alone cannot properly show signs of any semantic adaptation to the task at hand.

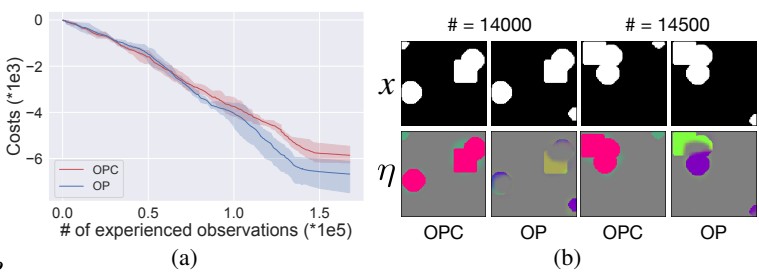

*o* (a) (b)

On the other hand, the quality and consistency of the soft-assignment generated by OPC are improved with the TD signal, suggesting that RL facilitates the learning of the perception model. Furthermore, the interaction between objects shown in Fig. (5b) demonstrates the agent is moving away from the poison (even when the food is nearby). Because of the

Figure 5: (a) Performance comparison between OPC and OP. (b) The soft-assignment $\eta$ after experiencing $14000$ and $14500$ observations. Each $\psi_k$ $^\eta$is assigned with a different color as in Fig. (4). $x$ is the binary-processed original observation.

environment setting of the high moving speed, larger target sizes and more poison objects, we believe that agents are given strong indications to stay away from all stimuli. Thus, the result shows an understanding of visual reasoning that the agent can separate itself from the rest of the objects.

### 4.5 ABLATION STUDY ON HYPER-PARAMETERS

We also investigate the effect of hyper-parameters to the learning of OPC, by changing: 1) the number of recurrent copies $K$, 2) the rollout steps for recurrent iteration $t_{ro}$, and 3) the use of relational mechanism across object representations $\boldsymbol{\theta}$ (see Sect. 2.2 of van Steenkiste *et al.* (2018) for more details). Fig. (3c) and Fig. (3d) show period reward results across different hyper-parameter settings.

As illustrated in Fig. (3c), the number of recurrent copies affects the stability of OPC learning, as OPC with $K = 3$ has experienced larger variance during the agent's life-cycle. We believe the difference comes from the environment dynamics as we have visually four objects in the environment. During the earlier stage of interacting with the environment, OPC tries to group each object into a distinct class; thus, a different number of $K$ against the number of objects in the environment confuse the perception model and lead to unstable learning. Although different $K$ settings might affect the learning stability and slow down the convergence, OPC can still find an optimal strategy within a given amount of observations.

In Fig. (3d), we compare OPC with different steps of recurrent rollout $t_{ro}$. A smaller $t_{ro}$ means fewer rounds of perception updates and therefore slower convergence in terms of the number of experienced observations. We believe that the choice of $t_{ro}$ depends on the difficulty of the environment, e.g., a smaller $t_{ro}$ can help to find the optimal strategy more quickly for simpler environments in terms of wall training time.

Meanwhile, results in Fig. (3c) and Fig. (3d) show that the use of a relational mechanism has limited impact on OPC, possibly because the objects can be well distinguished and perceived by their object-level inductive biases, i.e. shapes in our experiment. We believe that investigating whether the relational mechanism will have impact on environments where entities with similar object-level inductive bias have other different internal properties is an interesting direction for future work.

## 5 CONCLUSIONS

In this paper, we propose joint Perception and Control as Inference (PCI), a general framework to combine perception and control for POMDPs through Bayesian inference. We then extend PCI to the context of a typical pixel-level environment with compositional structure and propose Object-based Perception Control (OPC), an instantiation of PCI which manages to facilitate control with the help of automatically discovered object-based representations. We provide the convergence proof of OPC perception model update and demonstrate the execution-time optimization ability of OPC in a high-dimensional pixel environment. Notably, our experiments show that OPC achieves high quality and consistent perceptual grouping and outperforms several strong baselines in terms of accumulated rewards within the agent's life-cycle. OPC agent can quickly learn the dynamics of a new environment without any prior knowledge, imitating the inductive bias acquisition process of humans. For future work, we would like to investigate OPC with more types of inductive biases and test the model performance in a wider variety of environments.

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

## A   CONVERGENCE OF THE OBJECT-BASED PERCEPTION MODEL UPDATE

Under main assumptions and lemmas as introduced below, we demonstrate the convergence of a sequence of $\{\mathcal{L}^w_{\boldsymbol{\theta}^t}(q^w, \phi)\}$ generated by the perception update. The proof is presented by showing that the learning process follows the Global Convergence Theorem (Zangwill, 1969).

**Assumption 1.** $\Omega_{\boldsymbol{\theta}^0} = \{\boldsymbol{\theta} \in \Omega : \mathcal{L}^w_{\boldsymbol{\theta}}(q^w, \phi) \leq \mathcal{L}^w_{\boldsymbol{\theta}^0}(q^w, \phi)\}$ is compact for any $\mathcal{L}^w_{\boldsymbol{\theta}^0}(q^w, \phi) < \infty$.

**Assumption 2.** $\mathcal{L}$ is continuous in $\Omega$ and differentiable in the interior of $\Omega$.

The above assumptions lead to the fact that $\{\mathcal{L}^w_{\boldsymbol{\theta}^t}(q^w, \phi)\}$ is bounded for any $\boldsymbol{\theta}^0 \in \Omega$.

**Lemma 1.** Let $\Omega_S$ be the set of stationary points in the interior of $\Omega$, then the mapping $\arg\max_{\boldsymbol{\theta}^{t+1}} \Lambda_{\boldsymbol{\theta}^t}(\boldsymbol{\theta}^{t+1})$ from Eq. 4 is closed over $\Omega \backslash \Omega_S$ (the complement of $\Omega_S$).

*Proof.* See Wu (1983). A sufficient condition is that $\Lambda_{\boldsymbol{\theta}^t}(\boldsymbol{\theta}^{t+1})$ is continuous in both $\boldsymbol{\theta}^{t+1}$ and $\boldsymbol{\theta}^t$. □

**Proposition 1.** Let $\Omega_S$ be the set of stationary points in the interior of $\Omega$, then (i.) $\forall \boldsymbol{\theta}^t \in \Omega_S, \mathcal{L}^w_{\boldsymbol{\theta}^{t+1}}(q^w, \phi) \leq \mathcal{L}^w_{\boldsymbol{\theta}^t}(q^w, \phi)$ and (ii.) $\forall \boldsymbol{\theta}^t \in \Omega \backslash \Omega_S, \mathcal{L}^w_{\boldsymbol{\theta}^{t+1}}(q^w, \phi) < \mathcal{L}^w_{\boldsymbol{\theta}^t}(q^w, \phi)$.

*Proof.* Note that (i.) holds true given the condition. To prove (ii.), consider any $\boldsymbol{\theta}^t \in \Omega \backslash \Omega_S$, we have

$$\frac{\partial \mathcal{L}^w_{\boldsymbol{\theta}^t}(q^w, \phi)}{\partial \boldsymbol{\theta}^{t+1}}\bigg|_{\boldsymbol{\theta}^{t+1}=\boldsymbol{\theta}^t} = \frac{\partial \Lambda_{\boldsymbol{\theta}^t}(\boldsymbol{\theta}^{t+1})}{\partial \boldsymbol{\theta}^{t+1}}\bigg|_{\boldsymbol{\theta}^{t+1}=\boldsymbol{\theta}^t} \neq 0.$$

Hence $\Lambda_{\boldsymbol{\theta}^t}(\boldsymbol{\theta}^{t+1})$ is not maximized at $\boldsymbol{\theta}^{t+1} = \boldsymbol{\theta}^t$. Given the perception update described by Eq. (5), we therefore have $\Lambda_{\boldsymbol{\theta}^t}(\boldsymbol{\theta}^{t+1}) > \Lambda_{\boldsymbol{\theta}^t}(\boldsymbol{\theta}^t)$, which implies $\mathcal{L}^w_{\boldsymbol{\theta}^{t+1}}(q^w, \phi) < \mathcal{L}^w_{\boldsymbol{\theta}^t}(q^w, \phi)$. □

**Theorem 1.** Let $\{\boldsymbol{\theta}^t\}$ be a sequence generated by the mapping from Eq. 4, $\Omega_S$ be the set of stationary points in the interior of $\Omega$. If Assumptions 1 & 2, Lemma 1, and Proposition 1 are met, then all the limit points of $\{\boldsymbol{\theta}^t\}$ are stationary points (local minima) and $\mathcal{L}^w_{\boldsymbol{\theta}^t}(q^w, \phi)$ converges monotonically to $J^* = \mathcal{L}^w_{\boldsymbol{\theta}^*}(q^w, \phi)$ for some stationary point $\boldsymbol{\theta}^* \in \Omega_S$.

*Proof.* Suppose that $\boldsymbol{\theta}^*$ is a limit point of the sequence $\{\boldsymbol{\theta}^t\}$. Given Assumptions 1 & 2 and Proposition 1.i), we have that the sequence $\{\boldsymbol{\theta}^t\}$ are contained in a compact set $\Omega_K \subset \Omega$. Thus, there is a subsequence $\{\boldsymbol{\theta}^l\}_{l \in L}$ of $\{\boldsymbol{\theta}^t\}$ such that $\boldsymbol{\theta}^l \to \boldsymbol{\theta}^*$ as $l \to \infty$ and $l \in L$.

We first show that $\mathcal{L}^w_{\boldsymbol{\theta}^t}(q^w, \phi) \to \mathcal{L}^w_{\boldsymbol{\theta}^*}(q^w, \phi)$ as $t \to \infty$. Given $\mathcal{L}$ is continuous in $\Omega$ (Assumption 2), we have $\mathcal{L}^w_{\boldsymbol{\theta}^l}(q^w, \phi) \to \mathcal{L}^w_{\boldsymbol{\theta}^*}(q^w, \phi)$ as $l \to \infty$ and $l \in L$, which means

$$\forall \epsilon > 0, \exists l(\epsilon) \in L \text{ s.t. } \forall l \geq l(\epsilon), l \in L, \mathcal{L}^w_{\boldsymbol{\theta}^l}(q^w, \phi) - \mathcal{L}^w_{\boldsymbol{\theta}^*}(q^w, \phi) < \epsilon. \tag{7}$$

Given Proposition 1 and Eq. (4), $\mathcal{L}$ is therefore monotonically decreasing on the sequence $\{\boldsymbol{\theta}^t\}_{t=0}^\infty$, which gives

$$\forall t, \mathcal{L}^w_{\boldsymbol{\theta}^t}(q^w, \phi) - \mathcal{L}^w_{\boldsymbol{\theta}^*}(q^w, \phi) \geq 0. \tag{8}$$

Given Eq. (7), for any $t \geq l(\epsilon)$, we have

$$\mathcal{L}^w_{\boldsymbol{\theta}^t}(q^w, \phi) - \mathcal{L}^w_{\boldsymbol{\theta}^*}(q^w, \phi) = \underbrace{\mathcal{L}^w_{\boldsymbol{\theta}^t}(q^w, \phi) - \mathcal{L}^w_{\boldsymbol{\theta}^{l(\epsilon)}}(q^w, \phi)}_{\leq 0} + \underbrace{\mathcal{L}^w_{\boldsymbol{\theta}^{l(\epsilon)}}(q^w, \phi) - \mathcal{L}^w_{\boldsymbol{\theta}^*}(q^w, \phi)}_{< \epsilon} < \epsilon. \tag{9}$$

Given Eq. (8) and Eq. (9), we therefore have $\mathcal{L}^w_{\boldsymbol{\theta}^t}(q^w, \phi) \to \mathcal{L}^w_{\boldsymbol{\theta}^*}(q^w, \phi)$ as $t \to \infty$. We then prove that the limit point $\boldsymbol{\theta}^*$ is a stationary point. Suppose $\boldsymbol{\theta}^*$ is not a stationary point, i.e., $\boldsymbol{\theta}^* \in \Omega \backslash \Omega_S$, we consider the sub-sequence $\{\boldsymbol{\theta}^{l+1}\}_{l \in L}$, which are also contained in the compact set $\Omega_K$. Thus, there is a subsequence $\{\boldsymbol{\theta}^{l'+1}\}_{l' \in L'}$ of $\{\boldsymbol{\theta}^{l+1}\}_{l \in L}$ such that $\boldsymbol{\theta}^{l'+1} \to \boldsymbol{\theta}^{*'}$ as $l' \to \infty$ and $l' \in L'$, yielding $\mathcal{L}^w_{\boldsymbol{\theta}^{l'+1}}(q^w, \phi) \to \mathcal{L}^w_{\boldsymbol{\theta}^{*'}}(q^w, \phi)$ as $l' \to \infty$ and $l' \in L'$, which gives

$$\mathcal{L}^w_{\boldsymbol{\theta}^{*'}}(q^w, \phi) = \lim_{\substack{l' \to \infty \\ l' \in L'}} \mathcal{L}^w_{\boldsymbol{\theta}^{l'+1}}(q^w, \phi) = \lim_{\substack{t \to \infty \\ t \in \mathbb{N}}} \mathcal{L}^w_{\boldsymbol{\theta}^t}(q^w, \phi) = \mathcal{L}^w_{\boldsymbol{\theta}^*}(q^w, \phi). \tag{10}$$

On the other hand, since the mapping from Eq. (4) is closed over $\Omega \backslash \Omega_S$ (Lemma 1), and $\boldsymbol{\theta}^* \in \Omega \backslash \Omega_S$, we therefore have $\boldsymbol{\theta}^{*'} \in \arg\max_{\boldsymbol{\theta}^{t+1}} \Lambda_{\boldsymbol{\theta}^*}(\boldsymbol{\theta}^{t+1})$, yielding $\mathcal{L}^w_{\boldsymbol{\theta}^{*'}}(q^w, \phi) < \mathcal{L}^w_{\boldsymbol{\theta}^*}(q^w, \phi)$ (Proposition 1.ii), which contradicts Eq. (10). □

## B EXPERIMENT DETAILS

**OPC**  In all experiments we trained the perception model using ADAM Kingma and Ba (2014) with default parameters and a batch size of 32. Each input consists of a sequence of binary $84 \times 84$ images containing two poison objects (two circles) and one food object (a rectangle) that start in random positions and move within the image for $t_{ro}$ steps. These frames were thresholded at $0.0001$ to obtain binary images and added with bit-flip noise ($p = 0.2$). We used a convolutional encoder-decoder architecture inspired by recent GANs Chen *et al.* (2016) with a recurrent neural network as bottleneck, where the encoder used the same network architecture from Mnih *et al.* (2013) as

1. $8 \times 8$ conv. 16 ELU. stride 4. layer norm
2. $4 \times 4$ conv. 32 ELU. stride 2. layer norm
3. fully connected. 256 ELU. layer norm
4. recurrent. 250 Sigmoid. layer norm on the output
5. fully connected. 256 RELU. layer norm
6. fully connected. $10 \times 10 \times 32$ RELU. layer norm
7. $4 \times 4$ reshape 2 nearest-neighbour, conv. 16 RELU. layer norm
8. $8 \times 8$ reshape 4 nearest-neighbour, conv. 1 Sigmoid

We used the Advantage Actor-Critic (A2C) Mnih *et al.* (2016) with an MLP policy as the decision making module of OPC. The MLP policy added a 512-unit fully connected layer with rectifier nonlinearity after layer 4 of the perception model. The decision making module had two set of outputs: 1) a softmax output with one entry per action representing the probability of selecting the action, and 2) a single linear output representing the value function. The decision making module was trained using RMSProp Tieleman and Hinton (2012) with a learning rate of $7e - 4$, a reward discount factor $\gamma = 0.99$, an RMSProp decay factor of $0.99$, and performed updates after every 5 actions.

**A2C**  We used the same convolutional architecture as the encoder of the perception model of OPC (layer 1 to 3), followed by a fully connected layer with 512 hidden units followed by a rectifier nonlinearity. The A2C was trained using the same setting as the decision making module of OPC.

**WM-A2C**  We used the same setting as Ha and Schmidhuber (2018) to separately train the V model and the M model. The experience was generated off-line by a random policy operating in the Pixel Waterworld environment. We concatenated the output of the V model and the M model as the A2C input, and trained A2C using the same setting as introduced above.

## C DETAILS OF DERIVATION

### C.1 DERIVATION OF EQ. (1)

$$\log p(o^{\geq t}, \boldsymbol{x}^{\leq t}|a^{<t})$$

$$= \log \sum_{\boldsymbol{s}^{\geq 1}, a^{\geq t}, \boldsymbol{x}^{>t}} p(o^{\geq t}, \boldsymbol{s}^{\geq 1}, \boldsymbol{x}^{\geq 1}, a^{\geq t}|a^{<t})$$

$$= \log \sum_{\boldsymbol{s}^{\geq 1}, a^{\geq t}, \boldsymbol{x}^{>t}} p(o^{\geq t}|\boldsymbol{s}^{\geq 1}, \boldsymbol{x}^{\geq 1}, a^{\geq 1}) p(\boldsymbol{s}^{\geq 1}, \boldsymbol{x}^{\geq 1}, a^{\geq 1}|a^{<t})$$

$$= \log \sum_{\boldsymbol{s}^{\geq 1}, a^{\geq t}, \boldsymbol{x}^{>t}} \prod_{n=t} p(o^n|\boldsymbol{s}^n, a^n) p(\boldsymbol{x}^{\leq t}|\boldsymbol{s}^{\geq 1}, \boldsymbol{x}^{>t}, a^{\geq 1}) p(\boldsymbol{s}^{\geq 1}, \boldsymbol{x}^{>t}, a^{\geq 1}|a^{<t})$$

$$= \log \sum_{\boldsymbol{s}^{\geq 1}, a^{\geq t}, \boldsymbol{x}^{>t}} \prod_{n=t} p(o^n|\boldsymbol{s}^n, a^n) \prod_{j=1}^{t} p(\boldsymbol{x}^j|\boldsymbol{s}^j) p(\boldsymbol{x}^{>t}|\boldsymbol{s}^{\geq 1}, a^{\geq 1}) p(\boldsymbol{s}^{\geq 1}, a^{\geq 1}|a^{<t})$$

$$= \log \sum_{\boldsymbol{s}^{\geq 1}, a^{\geq t}, \boldsymbol{x}^{>t}} \prod_{n=t} p(o^n|\boldsymbol{s}^n, a^n) \prod_{j=1}^{t} p(\boldsymbol{x}^j|\boldsymbol{s}^j) \prod_{k=t+1} p(\boldsymbol{x}^k|\boldsymbol{s}^k) p(\boldsymbol{s}^{>t}|\boldsymbol{s}^{\leq t}, a^{\geq 1}) p(\boldsymbol{s}^{\leq t}, a^{\geq 1}|a^{<t})$$

$$= \log \sum_{\boldsymbol{s}^{\geq 1}, a^{\geq t}, \boldsymbol{x}^{>t}} \prod_{n=t} p(o^n|\boldsymbol{s}^n, a^n) \prod_{j=1}^{t} p(\boldsymbol{x}^j|\boldsymbol{s}^j) \prod_{k=t+1} p(\boldsymbol{x}^k|\boldsymbol{s}^k) \prod_{l=t} p(\boldsymbol{s}^{l+1}|\boldsymbol{s}^l, a^l) p(\boldsymbol{s}^1) \prod_{m=1}^{t-1} p(\boldsymbol{s}^{m+1}|\boldsymbol{s}^m, a^m) \prod_{h=t} p(a^h|a^{<t})$$

$$= \log \left[ \sum_{\boldsymbol{s}^{\leq t}} \prod_{j=1}^{t} p(\boldsymbol{x}^j|\boldsymbol{s}^j) p(\boldsymbol{s}^1) \prod_{m=1}^{t-1} p(\boldsymbol{s}^{m+1}|\boldsymbol{s}^m, a^m) \left[ \underbrace{\sum_{\boldsymbol{s}^{>t}, a^{\geq t}, \boldsymbol{x}^{>t}} \prod_{n=t} p(o^n|\boldsymbol{s}^n, a^n) \prod_{k=t} p(\boldsymbol{x}^{k+1}|\boldsymbol{s}^{k+1}) p(\boldsymbol{s}^{k+1}|\boldsymbol{s}^k, a^k) p(a^k|a^{<t})}_{①} \right] \right]$$

$$= \log \left[ \sum_{\boldsymbol{s}^{\leq t}} q(\boldsymbol{s}^{\leq t}|\boldsymbol{x}^{\leq t}, a^{<t}) \frac{\prod_{j=1}^{t} p(\boldsymbol{x}^j|\boldsymbol{s}^j) p(\boldsymbol{s}^1) \prod_{m=1}^{t-1} p(\boldsymbol{s}^{m+1}|\boldsymbol{s}^m, a^m)}{q(\boldsymbol{s}^{\leq t}|\boldsymbol{x}^{\leq t}, a^{<t})} [①] \right]$$

### C.2 DERIVATION OF EQ. (4)

$$\boldsymbol{\theta}^{t+1} = \arg\max_{\boldsymbol{\theta}^{t+1}} q^w \log \frac{\prod_{j=1}^{t} p(\boldsymbol{x}^j|\boldsymbol{s}^j) p(\boldsymbol{s}^1) \prod_{m=1}^{t-1} p(\boldsymbol{s}^{m+1}|\boldsymbol{s}^m, a^m)}{p(\boldsymbol{s}^1) \prod_{g=1}^{t-1} q(\boldsymbol{s}^{g+1}|\boldsymbol{x}^{\leq g}, a^{\leq g})}$$

$$= \arg\max_{\boldsymbol{\theta}^{t+1}} q^w \log \frac{p(\boldsymbol{s}^1) p(\boldsymbol{x}^1|\boldsymbol{s}^1) \prod_{j=1}^{t-1} p(\boldsymbol{x}^{j+1}, \boldsymbol{s}^{j+1}|\boldsymbol{s}^j, a^j)}{p(\boldsymbol{s}^1) \prod_{g=1}^{t-1} q(\boldsymbol{s}^{g+1}|\boldsymbol{x}^{\leq g}, a^{\leq g})}$$

$$= \arg\max_{\boldsymbol{\theta}^{t+1}} \eta^t \log \frac{p(\boldsymbol{x}^1|\boldsymbol{s}^1) \prod_{j=1}^{t-1} p(\boldsymbol{x}^{j+1}, \boldsymbol{s}^{j+1}|\boldsymbol{s}^j, a^j)}{\prod_{g=1}^{t-1} \eta^{g+1}} \text{ drop terms which are constant with respect to } \boldsymbol{\theta}^{t+1}$$

$$= \arg\max_{\boldsymbol{\theta}^{t+1}} \mathbb{E}_{\boldsymbol{z}^{t+1} \sim \eta^t} \left[ \log p_\phi(\boldsymbol{x}^{t+1}, \boldsymbol{z}^{t+1}, \boldsymbol{\psi}^{t+1}|\boldsymbol{s}^t, a^t) \right]$$

$$\doteq \arg\max_{\boldsymbol{\theta}^{t+1}} \Lambda_{\boldsymbol{\theta}^t}(\boldsymbol{\theta}^{t+1})$$

## C.3 Derivation of Eq. (5)

$$\frac{\partial \Lambda_{\boldsymbol{\theta}^{t+1}}(\boldsymbol{\theta}^{t+2})}{\partial \boldsymbol{\theta}_k^{t+1}} = \frac{\partial \Lambda_{\boldsymbol{\theta}^{t+1}}(\boldsymbol{\theta}^{t+2})}{\partial \boldsymbol{\psi}_k^{t+1}} \cdot \frac{\partial \{\boldsymbol{\psi}_k^{t+1}\}^T}{\partial \{\boldsymbol{\theta}_k^{t+1}\}^T}$$

$$= \left[ \frac{\partial \Lambda_{\boldsymbol{\theta}^{t+1}}(\boldsymbol{\theta}^{t+2})}{\partial \psi_{1,k}^{t+1}} \cdots \frac{\partial \Lambda_{\boldsymbol{\theta}^{t+1}}(\boldsymbol{\theta}^{t+2})}{\partial \psi_{D,k}^{t+1}} \right] \cdot \begin{bmatrix} \frac{\partial \psi_{1,k}^{t+1}}{\partial \boldsymbol{\theta}_k^{t+1}} \\ \vdots \\ \frac{\partial \psi_{D,k}^{t+1}}{\partial \boldsymbol{\theta}_k^{t+1}} \end{bmatrix}$$

$$= \sum_{i=1}^D \frac{\partial \Lambda_{\boldsymbol{\theta}^{t+1}}(\boldsymbol{\theta}^{t+2})}{\partial \psi_{i,k}^{t+1}} \cdot \frac{\partial \psi_{i,k}^{t+1}}{\partial \boldsymbol{\theta}_k^{t+1}}$$

$$= \sum_{i=1}^D \frac{\partial}{\partial \psi_{i,k}^{t+1}} \left[ \sum_{\boldsymbol{z}_i^{t+1}} p_\phi(\boldsymbol{z}_i^{t+1}|x_i^{t+1}, \boldsymbol{\psi}_i^t, a^t) \log p_{\boldsymbol{\psi}_i^{t+1}}(x_i^{t+1}, \boldsymbol{z}_i^{t+1}|\boldsymbol{s}^t, a^t) \right] \cdot \frac{\partial \psi_{i,k}^{t+1}}{\partial \boldsymbol{\theta}_k^{t+1}}$$

$$= \sum_{i=1}^D \frac{\partial}{\partial \psi_{i,k}^{t+1}} \left[ \sum_{\boldsymbol{z}_i^{t+1}} p_\phi(\boldsymbol{z}_i^{t+1}|x_i^{t+1}, \boldsymbol{\psi}_i^t, a^t) \log p_{\boldsymbol{\psi}_i^{t+1}}(x_i^{t+1}|\boldsymbol{z}_i^{t+1}) p(\boldsymbol{z}_i^{t+1}|\boldsymbol{s}^t, a^t) \right] \cdot \frac{\partial \psi_{i,k}^{t+1}}{\partial \boldsymbol{\theta}_k^{t+1}}$$

$$= \sum_{i=1}^D \frac{\partial}{\partial \psi_{i,k}^{t+1}} \left[ \sum_{k=1}^K p_\phi(z_{i,k}^{t+1}=1|x_i^{t+1}, \psi_{i,k}^t, a^t) \log p_{\psi_{i,k}^{t+1}}(x_i^{t+1}|\boldsymbol{z}_i^{t+1}) p(\boldsymbol{z}_i^{t+1}|\boldsymbol{s}^t, a^t) \right] \cdot \frac{\partial \psi_{i,k}^{t+1}}{\partial \boldsymbol{\theta}_k^{t+1}}$$

$$= \sum_{i=1}^D p_\phi(z_{i,k}^{t+1}=1|x_i^{t+1}, \boldsymbol{\psi}_i^t, a^t) \frac{\partial}{\partial \psi_{i,k}^{t+1}} \left[ \log p_{\psi_{i,k}^{t+1}}(x_i^{t+1}|\boldsymbol{z}_i^{t+1}) p(\boldsymbol{z}_i^{t+1}|\boldsymbol{s}^t, a^t) \right] \cdot \frac{\partial \psi_{i,k}^{t+1}}{\partial \boldsymbol{\theta}_k^{t+1}}$$

$$= \sum_{i=1}^D p_\phi(z_{i,k}^{t+1}=1|x_i^{t+1}, \boldsymbol{\psi}_i^t, a^t) \frac{\partial}{\partial \psi_{i,k}^{t+1}} \left[ \log p_{\psi_{i,k}^{t+1}}(x_i^{t+1}|\boldsymbol{z}_i^{t+1}) + \log p(\boldsymbol{z}_i^{t+1}|\boldsymbol{s}^t, a^t) \right] \cdot \frac{\partial \psi_{i,k}^{t+1}}{\partial \boldsymbol{\theta}_k^{t+1}}$$

$$= \sum_{i=1}^D p_\phi(z_{i,k}^{t+1}=1|x_i^{t+1}, \boldsymbol{\psi}_i^t, a^t) \frac{\partial}{\partial \psi_{i,k}^{t+1}} \left[ \log p_{\psi_{i,k}^{t+1}}(x_i^{t+1}|\boldsymbol{z}_i^{t+1}) \right] \cdot \frac{\partial \psi_{i,k}^{t+1}}{\partial \boldsymbol{\theta}_k^{t+1}}$$

$$= \sum_{i=1}^D p_\phi(z_{i,k}^{t+1}=1|x_i^{t+1}, \boldsymbol{\psi}_i^t, a^t) \frac{\partial}{\partial \psi_{i,k}^{t+1}} \left[ \log \left[ \frac{1}{\sqrt{2\pi\sigma^2}} \exp\left( -\frac{1}{2} \frac{(x_i^{t+1} - \psi_{i,k}^{t+1})^2}{\sigma^2} \right) \right] \right] \cdot \frac{\partial \psi_{i,k}^{t+1}}{\partial \boldsymbol{\theta}_k^{t+1}}$$

$$= \sum_{i=1}^D p_\phi(z_{i,k}^{t+1}=1|x_i^{t+1}, \boldsymbol{\psi}_i^t, a^t) \frac{\partial}{\partial \psi_{i,k}^{t+1}} \left[ -\log \sigma - \log \sqrt{2\pi} - \frac{1}{2} \frac{(x_i^{t+1} - \psi_{i,k}^{t+1})^2}{\sigma^2} \right] \cdot \frac{\partial \psi_{i,k}^{t+1}}{\partial \boldsymbol{\theta}_k^{t+1}}$$

$$= \sum_{i=1}^D p_\phi(z_{i,k}^{t+1}=1|x_i^{t+1}, \boldsymbol{\psi}_i^t, a^t) \cdot -\frac{1}{2} \frac{2(x_i^{t+1} - \psi_{i,k}^{t+1}) \cdot (-1)}{\sigma^2} \cdot \frac{\partial \psi_{i,k}^{t+1}}{\partial \boldsymbol{\theta}_k^{t+1}}$$

$$\overset{Eq.~(3)}{=} \sum_{i=1}^D \eta_{i,k}^t \cdot \frac{\psi_{i,k}^{t+1} - x_i^{t+1}}{\sigma^2} \cdot \frac{\partial \psi_{i,k}^{t+1}}{\partial \boldsymbol{\theta}_k^{t+1}}.$$

