# OpenReview forum: "Joint Perception and Control as Inference with an Object-based Implementation"
_ICLR.cc/2021/Conference — Reject_

### Official Review · AnonReviewer3 · 2020-10-28
**Perception and control in one objective, but lacking empirical support and clean presentation of main results**

**Rating:** 4
**Confidence:** 3

**Review:**

This paper features two complementary contributions:
1.  The perception and control as inference (PCI) framework, which describes the graphical model of a POMDP with auxiliary optimality variables, allowing for the derivation of an objective for optimizing both a perception model and policy jointly to maximize $\log p(\mathcal{O}, \mathbf{x} \mid \mathbf{a})$. As the name suggests, this is similar to the control as inference formulation with a focus on partial observability and the addition of a perception model $q^w$.
2. Object-based perception control (OPC), which is an instance of PCI using an object-factorized model.

Incorporating a perception model into the the control-as-inference derivation to yield a single unified objective for multiple components in a pipeline is an interesting idea and complements other work on incorporating reward structure into model-learning nicely. (A discussion of some of these, such as [[Farahmand et al, 2017]](http://proceedings.mlr.press/v54/farahmand17a.html) or [[Oh et al, 2017]](https://arxiv.org/abs/1707.03497) could serve to better highlight the difference between PCI and existing hybrid approaches, but these references are not "missing" in the sense that their absence is an issue.)

However, the experimental evaluation does not really do the generality of this idea justice. It seems that this sort of approach would be favored only when the perception problem is too difficult for a standard maximum likelihood objective, as the joint objective would (presumably) bias the perception model to be useful for control. The main evaluation of the perception model, though, only shows four primitive white shapes on a black background being assigned labels. Unless there is underlying complexity not apparent in the image, this sort of input appears to be in line with what binary segmentation algorithms could work with, and likely would require no learning. The implicit argument made is that optimizing the joint objective causes these labels to be semantically-informed, but this is not supported by an experiment; the segmentation results in figure 4 are also what you would get by segmenting based on boundaries.

Somewhat better support could come from showing what labels are assigned when two different object types are contiguous with each other (would food and poison be assigned the same label, or does the model separate them?), but in general it is difficult to support the claimed benefits with this particular environment. This is likely one of the reasons for the result described in the figure 3(c-d) ablations: when the environment is too simple (this time in terms of dynamics, likely, and not perceptual complexity), many of the design choices can become less consequential.

Aside from experiments, the presentation of the core ideas could also be improved somewhat. Section 3.2 in particular is difficult to parse, and presented in such a way that many readers will likely just skip over most of it. It is certainly a good idea to work out an approach in its full generality, but some of the components taking up space in the presented derivation are immediately cut in the experiments section (like the $D_\text{KL}(\pi~ || ~p(a^t \mid a^{<t})$ term), so you might as well present a simplified version of the result that focuses on the method actually used.


**Miscellaneous questions**
1. What is the role of $p(a \mid a^{<t})$? Is this to account for a recursive policy?
2. Why do you think the world models baseline never appreciably improves upon its random initialization in figure 2b? I could imagine that the joint objective would cause OPC to perform a bit better than world models at convergence, but I am surprised that the perception is apparently so difficult that a separate modeling objective never gets off the ground at all.


**Minor**
1. Section 3.4: "resulting a" $\rightarrow$ "resulting in a"
2. There are a number of claims about human cognition that are not supported. For example:

> OPC agent can quickly learn the dynamics of a new environment without any prior knowledge, imitating the inductive bias acquisition process of humans.

> Moreover, in order to mimic a human’s spontaneous acquisition of inductive biases throughout its life, we propose to build a model able to acquire new knowledge online, rather than a one which merely generates static information from offline training (Dehaene et al., 2017).

It is hard to say whether OPC mimics humans' knowledge acquisition without any sort of evidence.


**Summary**

This generalization of the control as inference formulation could be influential given a more didactic presentation of the PCI derivation and an evaluation better suited to its claimed strengths, but it does not seem quite ready for publication yet. I would encourage the authors to try to test their approach on an environment that is perceptually complicated enough to motivate a non-standard perceptual modeling objective.

---

> ### Author Response · Authors · 2020-11-25
> **Response to Reviewer3**
>
> Thank you for your valuable feedback.
>
> **On additional related work discussion.**
> Please see the general response.
>
> **On the benefit of joint inference.**
> Please see the general response and Sect. 4.4 of the revised version.
>
> **On simpler Sec 3.2.**
> We really appreciate the reviewer's understanding of our intention for providing a framework with full generality in Section 3.2.
> During the time we wrote the paper and after receiving your valuable reviews, we had rounds of discussions about this issue within our group.
> We tend to maintain its full generality in this section so that we and other readers of this paper could easily take it as a starting point for various modifications and improvements in future works.
> We also worry that a simplified version in the section where the framework is first formally introduced could cause potential confusion for readers who try to verify and derive the PCI framework by hand.
>
>
> **On the role of $p(a \mid a^{<t})$.**
>
> This $p(a \mid a^{<t})$ conditional probability of the current action conditioned on previous actions was derived from a joint probability $p(s^{\le t}, a^{>t}  \mid a^{<t})$ by the Bayes rule.
> It serves as a prior in the variational bound we provided.
> In our experiments, we do not account for a recursive policy, and therefore, we explicitly assume that $p(a \mid a^{<t}) = p(a)$.
> We did not simplify the equation by this way as we wanted to keep the generality of this framework so future works could reuse it when requiring or assuming a recursive policy.

---

### Official Review · AnonReviewer2 · 2020-10-28
**Interesting application of unsupervised object segmentation within control as inference for POMDPs, but needs better framing**

**Rating:** 5
**Confidence:** 4

**Review:**

Summary
-------------
The main contribution of the paper is the incorporation of the object-based inference module (Greff et al., 2017; van Steenkiste et al., 2018) into the control-as-inference framework under partial observability. The method is evaluated on a version of Waterworld environment.

Decision
-----------
I think the paper is borderline, and I tend towards rejecting it in its current form. But I am open to increase my rating if the author address my concerns. Below are the main reasons for my decision.

1. The separation between what is novel and what is already known is not clear from the paper. For example, it is claimed that the paper introduces a novel framework, however there is a number of papers working with the same framework but using different approximations, e.g., [1,2,3]. There is also a close relation to active inference papers, e.g., [4,5]. In my opinion, it should be stated more clearly in the paper that the main contribution is the application of the object-based perception model within the control-as-inference framework, and not an introduction of a novel framework.
2. Following from the previous point, since the perception module is the main innovation, its evaluation seems insufficient. It is tested only on one environment. Would it generalize to other environments?

References
---------------
[1] Ortega, P. A., Wang, J. X., Rowland, M., Genewein, T., Kurth-Nelson, Z., Pascanu, R., ... & Jayakumar, S. M. (2019). Meta-learning of sequential strategies. arXiv preprint arXiv:1905.03030.
[2] Tschiatschek, S., Arulkumaran, K., Stühmer, J., & Hofmann, K. (2018). Variational inference for data-efficient model learning in pomdps. arXiv preprint arXiv:1805.09281.
[3] O'Donoghue, B., Osband, I., & Ionescu, C. (2020). Making sense of reinforcement learning and probabilistic inference. arXiv preprint arXiv:2001.00805.
[4] Tschantz, A., Millidge, B., Seth, A. K., & Buckley, C. L. (2020). Reinforcement Learning through Active Inference. arXiv preprint arXiv:2002.12636.
[5] Friston, K., Da Costa, L., Hafner, D., Hesp, C., & Parr, T. (2020). Sophisticated Inference. arXiv preprint arXiv:2006.04120.

---

> ### Author Response · Authors · 2020-11-25
> **Response to Reviewer 2**
>
> Thank you for your valuable feedback.
>
> **On the novelty of PCI.**
> We thank the reviewer for providing additional references on formulating RL as inference and have updated the related work section in the revised version.
> A key difference of our work comparing to [1,2,3] is that we focus on the joint perception and control as inference to solving POMDPs,
> whereas [1] studied RL as inference without explicit modeling of the transition function and the observation distribution and [3] studied RL as inference for MDPs.
> [2] focuses on learning a model of POMDP by variational inference and then solve the POMDP by model-based planning with the model learned. However, we do not separate the learning of model and policy but treat them as a whole and join them with a unified inference framework.
>
> **On more experiments.**
> We agree that the PCI framework is general and can be applied to a broad range of problems.
> In this paper, we choose to focus on an intuitive use-case to serve as a proof of concept for joint perception and control as inference.
> We also point out that pixel waterworld requires agent to acquire new knowledge online,
> which applies more naturally to our demonstrated settings of simulating a human's spontaneous acquisition of inductive biases throughout its life.
> In the revised version,
> we have added additional evaluations on comparing OPC against OP,
> a perception model with the same architecture as OPC but no guidance of the TD signal from RL.
>
>
> **Reference**
>
> [1] Pedro A. Ortega and J. X. Wang and M. Rowland and Tim Genewein and Zeb Kurth-Nelson and Razvan Pascanu and N. Heess and J. Veness and A. Pritzel and P. Sprechmann and Siddhant M. Jayakumar and T. McGrath and K. Miller and Mohammad Gheshlaghi Azar and Ian Osband and Neil C. Rabinowitz and A. Gy{\"o}rgy and S. Chiappa and Simon Osindero and Y. Teh and H. V. Hasselt and N. D. Freitas and M. Botvinick and S. Legg. Meta-learning of sequential strategies. ArXiv, abs/1905.03030, 2019.
>
> [2] Sebastian Tschiatschek, Kai Arulkumaran, Jan Stuhmer, and Katja Hofmann. Variational inference for data-efficient model learning in pomdps. arXiv preprint arXiv:1805.09281, 2018.
>
> [3] Brendan O'Donoghue, Ian Osband, and Catalin Ionescu. Making sense of reinforcement learning and probabilistic inference. In International Conference on Learning Representations, 2020.

---

### Official Review · AnonReviewer1 · 2020-10-28
**Official Blind Review #1**

**Rating:** 4
**Confidence:** 4

**Review:**

This paper proposes an extension of the RL as Inference framework, and demonstrates how to use it to express an object-centric RL model and train it on simple environments. It appears to be a combination of NEM [1] with a simple TD-learning objective on top. Results are a bit hard to interpret but seem promising.

I’m quite conflicted about this paper.
On the one hand, I find the problem that they’re trying to tackle (i.e. object-centric RL) to be interesting, important and rather current. Their approach is also interesting, theoretically guided and sound.
However on the other hand I found the choice of introducing yet another framework (which isn’t terribly useful or novel either, see below) to be counter-productive for the paper; the derivations to be slightly strenuous (and one can argue already quite known in the field and expressed in simpler fashion elsewhere, see [2]); and perhaps of unclear novelty (i.e. what is novel in the perception aspect compared to NEM?). Finally, the results aren’t extremely convincing and they do not compare to a model which is extremely related to their proposal (OP3 [3]).

Comments/questions:

1. I do not feel like the way the abstract and introduction argue for a general framework “Joint Perception and Control as Inference” is productive. As mentioned during the introduction, this isn’t a particularly novel framework (considering RL as inference, all the work done in temporal generative model for RL, or the rather new formulation by Hafner 2020 [4]) and it doesn’t appear like it provides a very interesting fit to the object-centric RL model proposed.
   1. The derivations of section 3.2 are heavy, introduce a lot of notation, but ultimately aren’t that complicated.
How do they differ from the similar derivations performed by Hafner et al, 2019’s Appendix F (see [2])?
   2. I understand that performing them for POMDP would be valuable, but then it would be more useful to address what is missing from existing methods and just show the required changes.
   3. I would argue that the paper would be stronger if you only expressed the object-centric RL derivation, and put it into context with NEM and OP3, to express what is different about it.
   4. I found the E and M steps derivations interesting, but they are effectively the same as shown in NEM [1]? Did I miss something? It would be more helpful to contrast them, instead of rederiving them.
   5. Finally, the novel aspect of the control objective isn’t actually used in practice, as a simple TD error method is used to train the policy, which others have done without taking a “RL as inference” stance.
2. The overall idea of the model is extremely close to what has been done in OP3 [3], where they combine sequential IODINE with policy heads to solve tasks while simultaneously learning an iterative inference generative model.
   1. This work should be cited and thoroughly discussed. It also should be a baseline in the best of situations.
3. Does the proposed model also inherit the limitations of NEM?
   1. For example, are you limited to black and white observations?
   2. How computationally expensive is it to represent $\frac{\partial\psi_k}{\partial \theta_k}$ with a neural network?
4. Figure 3 is relatively hard to interpret and quite noisy, which doesn’t make the results extremely promising?
   1. Starting with accumulated rewards makes it confusing, as one could be made to believe that the agents never successfully learn to perform the task and avoid poisons.
   2. “Period costs” should really be called “Episodic returns”, as this is really more standard.
   3. The variance is very high and would benefit from extra seeds.
5. The performance of the learnt generative model in Figure 4 does not seem particularly great, especially given the simplicity of the domain considered? The segmentations are fine, but the reconstructions seem quite too blurry compared to current SOTA methods.
6. Given the similarity of the environments, COBRA [5] might have been a better baseline than the relatively unrelated VAE / World Model baseline selected.
   1. I would also argue that COBRA should be cited in the “Model-based Deep Reinforcement Learning” as a closely related object-centric model-based RL method, it is more relevant than the World Model paper.

So overall, I’d recommend rewriting the paper to be more explicit about its contributions and differences to existing works, as this would make it much easier to follow.

* [1] Greff et al, 2017, https://arxiv.org/abs/1708.03498
* [2] Hafner et al, 2019, https://arxiv.org/abs/1811.04551
* [3] Veerapaneni et al, 2019, https://arxiv.org/abs/1910.12827
* [4] Hafner et al, 2020, https://arxiv.org/abs/2009.01791
* [5] Watters et al, 2019, https://arxiv.org/abs/1905.09275

---

> ### Author Response · Authors · 2020-11-25
> **Response to Reviewer 1**
>
> Thank you for your valuable feedback.
>
> **On the PCI comparison to Hafner et al, 2019.**
> As stated in Hafner et al, 2019's Appendix Section F, their bound is a variational bound for latent dynamics models, it does not include control as inference in their derivation as we do.
> Our notation for $o^t$ stands for optimality variable at time step $t$ rather than the observation variable in their setting.
> We apologize if reviewers find some notations in our derivation unnecessary, we would be grateful if the reviewers could point out which part we could remove or simplify so we can improve the readability of the paper.
>
> **On the PCI comparison to Hafner et al, 2020.**
> As discussed in *Connecting Perception and Control* of Section 2,
> we argue that Hafner et al, 2020 provides a common foundation from which a wide range of objectives can be derived but sacrifices the precision in terms of a partially observable setting.
> Environments in many real-world decision-making problems are only partially observable, which signifies the importance of MBRL methods to solving POMDPs.
> However, the relevant and integrated discussion is omitted in Hafner et al, 2020.
> In contrast,
> we focus on the joint perception and control as inference for POMDPs and provide a specialized joint-objective as well as a practical implementation.
>
> **On the comparison between OPC and NEM.**
> Although OPC is built upon a previous unsupervised object segmentation back-end (NEM),
> we explore one step forward by proposing a joint framework for perceptual grouping and decision-making.
> This could help an agent to discover structured objects from raw pixels so that it could better tackle its decision problems.
> We provide additional experiments in Sect. 4.4 of the revised version to demonstrate the difference between OPC and OP,
> a perception model using the same architecture as NEM without reward signal.
>
> **On the novel aspect of the control objective.**
> Please see the general response and Sect. 4.4 of the revised version.
>
> **On the naming of the axis label.**
> We use *period costs* instead of *episodic returns* because each experiment for the curve in Figure 3 is a single run of the pixel waterworld without interruption,
> i.e.,
> only one episode.
> Therefore,
> we present the accumulated reward given a fixed number of environment interactions ($2e4$ of experienced observations).
>
> **On the quality of reconstruction.**
> Our implementation of the reconstruction model is based on the unsupervised perceptual grouping method proposed by [1].
> However, using other advanced unsupervised object representations learning methods as the perceptual model (such as IODINE [2] and MONet [3]) is a straightforward extension.
>
> **On additional related work discussions.**
> Thank you for pointing this out and we have added the reference of OP3 in the revised version.
> Although several recent works have investigated the unsupervised object extraction for reinforcement learning (including OP3 and COBRA),
> our proposed OPC focuses on the joint framework for perceptual and inference as control.
> The current unsupervised perceptual grouping implementation of OPC is based on NEM [1] but can be easily extended to other advanced methods such as IODINE [2] (the one used by OP3).
>
> **Reference**
>
> [1] Greff, Klaus, Sjoerd Van Steenkiste, and J{\"u}rgen Schmidhuber. "Neural expectation maximization." In Advances in Neural Information Processing Systems, pp. 6691-6701. 2017.
>
> [2] Klaus Greff, RaphaÃ«l Lopez Kaufmann, Rishabh Kabra, Nick Watters, Christopher Burgess, Daniel Zoran, Loic Matthey, Matthew Botvinick, and Alexander Lerchner. "Multi-object representation learning with iterative variational inference." In ICML, pp. 2424-2433. 2019.
>
> [3] Christopher P. Burgess, Loic Matthey, Nicholas Watters, Rishabh Kabra, Irina Higgins, Matthew M Botvinick, and Alexander Lerchner. "Monet: Unsupervised scene decomposition and representation." ArXiv, abs/1901.11390, 2019.

---

### Official Review · AnonReviewer4 · 2020-11-02
**Interesting paper, but missing evidence and discussion**

**Rating:** 4
**Confidence:** 4

**Review:**

The authors propose a framework for joint perception and control as inference (PCI) to combine perception and control for the case of POMDPs. The authors particularly focus on the case of hidden perceptual states linked to small image observations, which are composed of pixels belonging to up to exactly one object each. Their main proposal is denoted as OPC, which stands for object based perception and control, which serves the purpose of automatically discovering objects from pixels while controlling the system.

In order to posit the OPC model, the authors adopt one of the various recent unsupervised grouping models which assign one of K latent object identities to each pixel and hence group/segment images into K objects.  The rest of the model is a common POMDP setting set up to perform control as inference via usage of latent optimality variables per timepoint.

One of the major contributions in the paper is a well worked-out joint inference derivation, which performs amortized inference in this shared model utilizing an RNN.

Experimentally, the authors verify their joint model on a waterworld environment, which consists of simple object shapes with semantics. They compare their model against baselines such as A2C with convolutional networks as feature extractors , A2C with the World Model (Ha et al) as a feature extractor (also based on convolutional networks) and a random policy.
They unsurprisingly beat these baselines.

Criticisms:
My main problem with the paper is that the authors do not show that the joint model does anything. If they use one of these perceptual grouping models as feature extractors without joint inference, would A2C work well? Is it that feature layer that buys performance or a deeper interactions through the joint model they propose such that the actions resolve uncertainty about objectness in a nontrivial way they would not if used in a pipeline that doe snot perform joint inference.
This is not demonstrated convincingly and is a huge missed opportunity.
I am left without a firm understanding if the highlighted contribution of joint inference benefits this model in any way.
In addition, all the baselines have 'monolithic' feature extractors , while pretty clearly object based feature extractors appear useful.
I would also want to see a baseline where objects are known and we understand what the rest of their framework would do.
In general I find that even though the experimental framework is extremely toy, the authors have left a lot on the table empirically to demonstrate the utility of their method.
Is the lengthy derivation of updates for the model worth doing if we don't collect this evidence?
What is the time complexity in terms of EM steps that are needed?
I am less concerned about the toy nature of the chosen experiment, as it would suffice to prove the concept the authors are after, if used sufficiently for empirical rigor.

A secondary criticism is the lack of discussion and comparison to systems such as the Schema networks (Schema Networks: Zero-shot Transfer with a Generative Causal Model of Intuitive Physics by Kansky et al) which has long ago proposed object based planning modules and has not used object based models in the loop (yet).
There may be more such papers in the related literature which are on that thread of combining RL with object based representations that have not found their way in the discussions here and clearly should.

Overall:
I consider this paper interesting and agree with the authors that the proposal of joint models is appealing and worthwhile investigating, but I would require much stronger evaluation to show that the joint aspect of OPC is driving the performance here, instead of just the object-based featurization. In its current form the paper feels interesting, but unfinished.
The lapses in scholarship are also important to fix in order to fairly slot the paper correctly into the wider field.
However, this paper is promising and I look forward to seeing it mature.

---

> ### Author Response · Authors · 2020-11-25
> **Response to Reviewer 4**
>
> Thank you for your valuable feedback.
>
> **On the benefit of joint inference.**
> Please see the general response and Sect. 4.4 of the revised version.
>
> **On the object-based baseline.**
> Please see the above response.
> After the additional experiment in Sect. 4.4,
> we can see that the lengthy derivation of updates for the model worth doing.
>
> **On discussion and comparison to systems such as the Schema networks.**
> Thank you for pointing this out and we have added reference of Schema networks in the revised version.
> A distinguishing feature of our work in relation to previous works in object-based RL (including Schema networks) is that we provide the decision-making process with object-based abstractions of high-dimensional observations in an *unsupervised* manner.

---

### Author Response · Authors · 2020-11-25
**General Response to Reviewers' Comments**

We thank all reviewers for their thoughtful feedback. We have posted a revision to incorporate some of your feedback and would like to address some of the common concerns here.

**On the benefit of joint inference/ novel aspect of the control objective.**
To gain insights into the benefit of the control objective for joint inference,
we further compare OPC against OP, a perception model with the same architecture as OPC but no guidance of the TD signal from RL,
i.e.,
using the same unsupervised object segmentation back-end as in NEM [1].
Please see Sect. 4.4 of the revised version for more details.

**On additional related work discussion.**
We provide an additional discussion on related work in Sect. 2 of the revised version.


**Reference**

[1] Greff, Klaus, Sjoerd Van Steenkiste, and J{\"u}rgen Schmidhuber. "Neural expectation maximization." In Advances in Neural Information Processing Systems, pp. 6691-6701. 2017.

---

### Decision · Program_Chairs · 2021-01-07
**Final Decision**

**Decision:**

Reject

**Comment:**

This paper introduces an object perception and control method for RL, derived from a control-as-inference formulation within a POMDP.  The paper provides a theoretical derivation and experiments where the proposed joint-inference approach outperforms baselines.

The discussion focussed on understanding the paper's contribution relative to prior work. The reviewers highlighted the similarities with earlier systems (R1, R2, R4), the unclear benefits of joint inference over independently trained modules in the experiments (R3), and the lack of clarity of the presentation (R1, R2, R3).  The authors responded to some of these criticisms, bolstering the paper with additional experiments to show the benefits of joint inference and increasing the discussion of related work.  The reviewers examined the revisions and rebuttal and found the paper still did not resolve all their original concerns.  Two limitations mentioned in the final phase of the discussion were the use of a single environment to evaluate the general framework, and continuing doubts on the contribution of joint inference mechanism to the measured performance.

Four knowledgeable reviewers indicate reject as their concerns were not adequately resolved.  The paper is therefore rejected.